# Thermal Treatment under Vacuum for Obtaining a Quenchant from Rapeseed Oil

Ana Maria Sivriu [1], Olga Valerica Sapunaru [1,2,*], Ancaelena Eliza Sterpu [2], Doinita-Roxana Cioroiu Tirpan [2], Timur Vasile Chis [2] and Tanase Dobre [1]

1   Chemical and Biochemical Engineering Department, University "Politehnica" of Bucharest, 1-7 Gheorghe Polizu Str., 011061 Bucharest, Romania; sivriu_ana@yahoo.com (A.M.S.); tghdobre@gmail.com (T.D.)
2   Chemistry and Chemical Engineering Department, Faculty of Applied Sciences and Engineering, Ovidius University of Constanta, 124 Mamaia Blvd., 900527 Constanta, Romania; asterpu@univ-ovidius.ro (A.E.S.); tirpanroxana@gmail.com (D.-R.C.T.); timur.chis@gmail.com (T.V.C.)
*   Correspondence: olga.sapunaru@365.univ-ovidius.ro; Tel.: +40-762520001

**Abstract:** The aim of this study was to improve the quality of a vegetable oil, having in view its use as a quenchant for metallic parts in aircrafts. A process of pyrolysis under vacuum was applied to obtain a bio-oil with reduced viscosity and good quenching properties. Preliminarily, the rapeseed oil was fast pyrolyzed at temperature in the range of 300–375 °C and absolute pressure of 1 μbar. Some results such as viscosity and yields of bio-oil were obtained with a narrowing of the temperature range between 300–320 °C, for further processing. Quenching tests with bio-oils on stainless steel 25CD4 showed cooling curves closer to those of the standard mineral oil (Castrol Iloquench$^{TM}$ 1), by comparing them with unprocessed vegetable oil. The hardness of the steel after treatment rose from 29–30 HRC to 43–45 HRC, in accordance with requirements (35–45 HRC). Therefore, the conclusion is that bio-oils obtained by pyrolysis under vacuum are good quenchant proceeds from this study.

**Keywords:** bio-oil; pyrolysis under vacuum; quenchant; cooling curves; hardness

## 1. Introduction

Quenching is a process of hardening metal parts by rapidly cooling down the uniformly heated piece. This thermal treatment avoids unwanted microstructural changes in material due to the slow cooling after quenching resulting in pieces without metallurgical distortion or stress.

Quenchants for blacksmithing are various (oils, gas, water, salts, brine, polymers), and their selection depends on a few factors: the steel type, the dimension of the piece to be hardened, desired properties after quenching [1]. Mineral oils tailored for this specific application are the most frequently used, but also vegetable oil (canola, olive, palm kernel oil) are attractive because they are good cheap quenchants and come from renewable resources [2].

During the last few decades, the alternative of vegetable oils to mineral oil as quenchant agents was extensively studied [3–6], not only for the final result, the mechanical properties of the testing specimen [7,8], but also for the wetting behavior [9] and chemical stability of the vegetable oil during repeated quenching cycles [10,11]. All the scientists concluded that vegetable oils constitute good replacements for mineral oils in this application, if methods for minimizing its oxidation instability are applied. The methods include expoxydation [12], hydrogenation [13], and esterification [14].

Extensive studies of Prabhu and Fernandes [9,15] on palm, coconut, sunflower, groundnut and castor oil as bio-quenchants showed little difference in surface wettability whence quench severity is comparable to conventional quench mineral oils.

In many studies, the inquiry went deeper, by calculating the heat transfer rates [3,7,16] from metal surface to the oil, in different stages of the quenching process, and by performing cooling curves analysis [17,18]. Moreover, by combining cooling curves with time-temperature-transformation (TTT) diagrams, it is possible to predict the variation of hardness via the quench factor analysis [16]. The studies [3,6,17] revealed heat transfer coefficients frequently superior for the vegetable oils, but a different aspect of the cooling curve during quenching, due to different physical-chemical properties of the oils. However, the results of quenching were similar in the end [8].

Cooling curves analyses performed on crude and processed soybean oils performed by Totten et al. [19] showed that these oils have similar cooling behavior and the vegetable oils have faster cooling rates at high temperatures than the reference mineral oil [3].

It was thought that higher viscosity of the vegetable oil could be a break in developing a high cooling rate in convective transfer [3]. Starting from this presumption, we questioned if a light pyrolysis of the vegetable oil would create a bio-oil from rapeseed oil with a lower viscosity than the raw and good oxidation stability. The pyrolysis under vacuum is our choice since it was demonstrated that it provides rapid pyrolysis, lowering of process temperature [20,21], short process time and low energy consumption [22]. This method has not been tested yet for this purpose, and the pyrolytic oil will be tested in quenching low-alloy steel parts of aircraft to successfully replace the mineral oil.

## 2. Materials and Methods

### 2.1. Materials and Analysis Methods

The original quench oil was a mineral one, Iloquench$^{TM}$ 1, and the raw oil used in this study was waste/frying rapeseed oil from a collection centre in Bucharest. Their characteristics are shown in Table 1.

**Table 1.** The characteristics of the waste rapeseed oil in comparison with the mineral oil.

| Characteristic | Unit | Rapeseed Oil | Iloquench$^{TM}$ 1 |
|---|---|---|---|
| Density at 15 °C | g/cm$^3$ | 0.910 | 0.870 |
| Flash point (Marcusson) | °C | 239 | >190 |
| Iodine number | g I$_2$/100 g sample | 5.8 | - |
| Kinematic visccosity at 40 °C | mm$^2$/s | 43 | 20 |
| Kinematic visccosity at 100 °C | mm$^2$/s | 12.8 | 5 |
| Cooling rate at 300 °C | °C/s | 6.13 | 6.18 |

The oils were characterized as follows: density at 15 °C by method ASTM 1250 with apparatus Anton Paar model DMA 4500, kinematic viscosity at 40 °C and 100 °C by Ubbelohde viscometer (method SR ISO 3104:2002) and flash point by Marcusson apparatus (method SR 5489:2008). Iodine number is determined by a volumetric method (Romanian standard STAS 315/74), as an indication of the oil stability at the oxidation.

The first thermal treatment of stainless steel test specimens, the heating, was performed in a brick oven with electrical resistance, in reducing the H$_2$ atmosphere. Then, the cooling rate at 300 °C was determined with the IVF Smart Quench apparatus, the same used to trace the cooling curves in quenching process.

The Wilson UH4250 apparatus was used to measure the Rockwell hardness of test bars before and after quenching.

### 2.2. Pyrolysis under Vacuum

Preliminary trials were performed in order to set the optimal temperature range. The pyrolysis took place in an electrical oven coupled with a vacuum system (Figure 1), fol-

lowed by cooling in argon 99.9% pure atmosphere to obtain the bio-oil. The test procedure is the following: the batch sample is put inside the oven, the oven door is closed tightly, the vacuum system is turned on and, after the work pressure is reached, the heating starts in accordance with the programme authomatically set: heating rate, temperature level, holding time at that level. When the time expired, the inert gas is admitted in the oven, the ventilator is turned on, thus ensuring the cooling the chamber and the bio-oil remaining in the batch is collected.

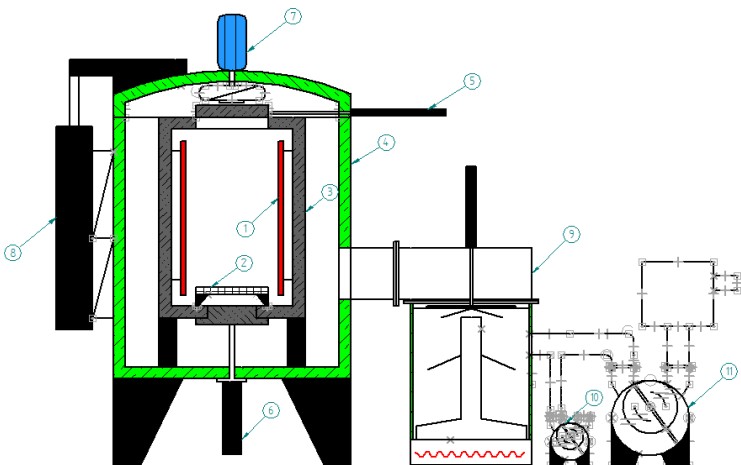

**Figure 1.** Experimental set for the pyrolysis under vacuum. Legend: 1—electrical resistance; 2—support for the batch; 3—heating chamber; 4—external jacket; 5—pneumatic cylinder for drive up; 6—pneumatic cylinder for drive down; 7—ventilator; 8—hydro-pneumatic cylinder for driving the oven's cap; 9—molecular diffusion pump; 10,11—vacuum pumps with eccentric drum and sliding paddle.

The trials were undertaken at absolute pressure of $1 \times 10^{-3}$ mbar at 300 °C and 375 °C respectively, for 20 min. Process watching was ensured by a recording instrument with one second resolution and six channels for tracking the temperature in the heating chamber, the temperature of the batch in three points and the pressure. In the end, data are automatically processed and represented graphically, as seen in Figures 2 and 3, for the trials at 300 °C and 375 °C.

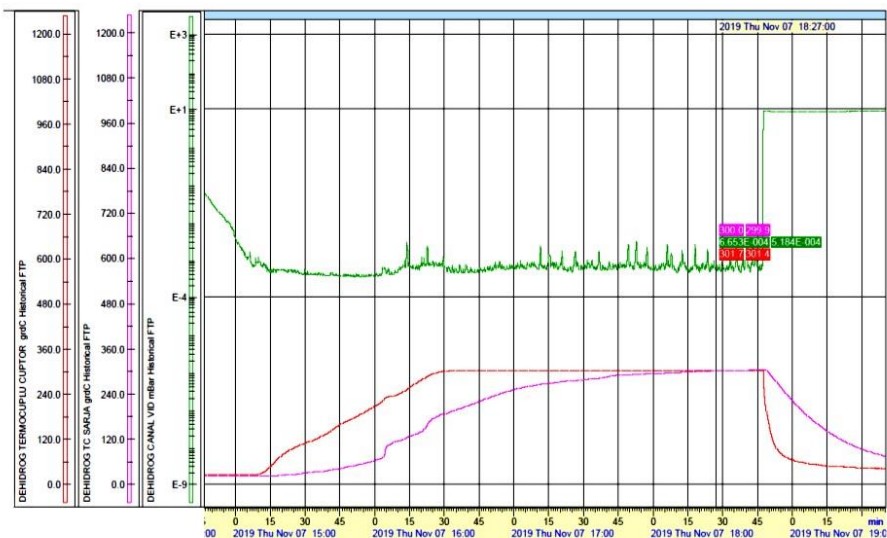

**Figure 2.** Process diagram for the pyrolysis under vacuum at 300 °C. Legend: ———— pressure; ———— batch temperature; ———— oven temperature.

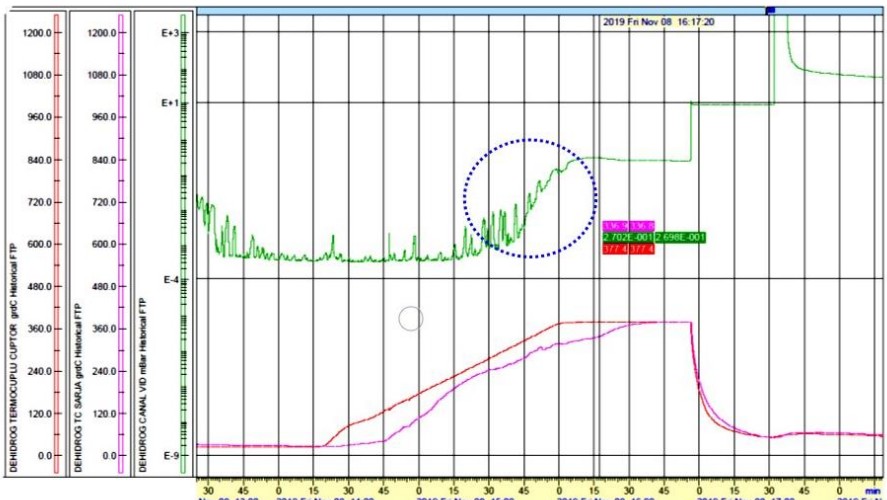

**Figure 3.** Process diagram for the pyrolysis under vacuum at 375 °C. Legend: ——— pressure; ——— batch temperature; ——— oven temperature.

As seen in Figures 2 and 3, there is a difference between the behaviour of the system at 300 °C and at 375 °C. Since at lower temperature the pressure is stable during the process, at 375 °C, due to the large quantity of gases formed and exceeding the capacity of the vacuum system, the pressure rises continuously. At 300 °C, the pressure was $1.79 \times 10^{-3}$ mbar, and at 375 °C was $1.97 \times 10^{-1}$ mbar and no liquid was left in the batch. Also, at 300 °C, the bio-oil yield was already 87%, so the pyrolysis should take place at temperatures close to 300 °C such that the yield does not decrease too much, and the next trials were undertaken at 310 °C and 320 °C, trying to obtain reasonable quantities of bio-oil. In Figures 4 and 5, one can see the diagrams for the pyrolysis at 310 °C and 320 °C.

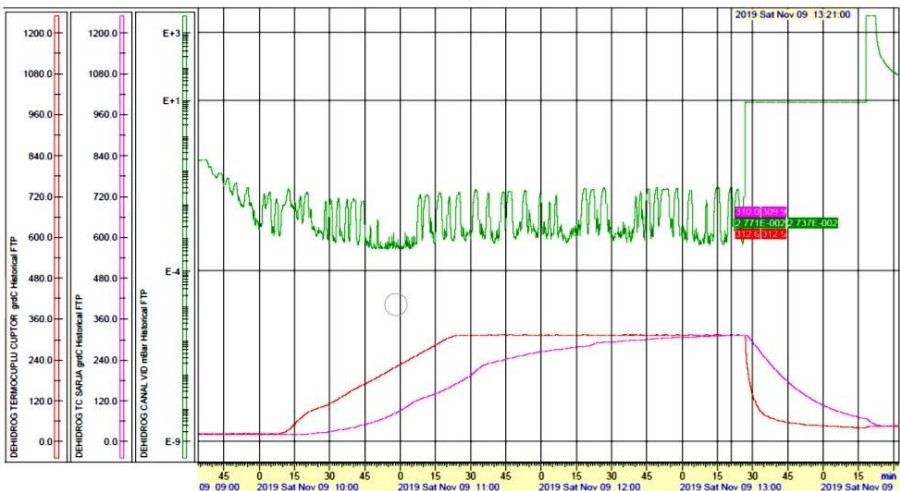

**Figure 4.** Process diagram for the pyrolysis under vacuum at 310 °C. Legend: ——— pressure; ——— batch temperature; ——— oven temperature.

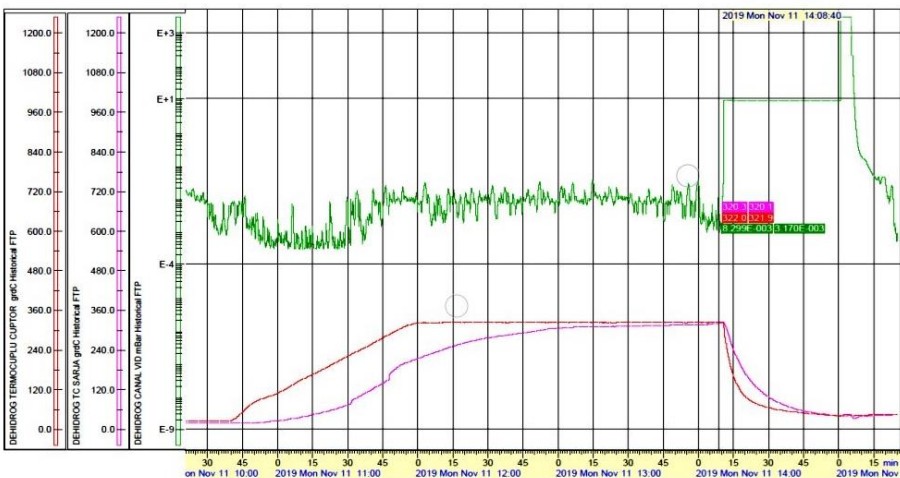

**Figure 5.** Process diagram for the pyrolysis under vacuum at 320 °C. Legend: ——— pressure; ——— batch temperature; ——— oven temperature.

As seen, at 310 °C and 320 °C, the pressure is maintained at the desired temperature level during the process at $1.8 \times 10^{-3}$ mbar and $1 \times 10^{-2}$ mbar, respectively.

Finally, the bio-oils were characterized by kinematic viscosity, flash point, iodine number and then by cooling curves.

### 2.3. Quenching Tests

The thermal treatment was performed on a set of test specimens made of stainless steel 25CD4 with the following composition: C—0.25 wt %, Si—0.25 wt %, Mn—0.7 wt %, Cr—1.05 wt %, Mo—0.25 wt %. The specimens were cylindrical, 8 mm dimetre and 300 mm length.

The heating of test specimens in the oven was undertaken at a controlled temperature of 850 °C ± 10 °C, maintaining this temperature for 30 min. Then, rapid cooling took place in the bio-oil. There were five tests: one for the mineral oil, one for rapeseed oil and three for the bio-oils resulting from the pyrolysis process at 300 °C, 310 °C and 320 °C, respectively.

The performance of the quenching process was measured as Rockwell hardness of the test samples.

### 3. Results and Discussion

The bio-oil quantities and yields after pyrolysis at 300 °C, 310 °C, 320 °C and 375 °C are found in Table 2.

**Table 2.** Bio-oil yields at pyrolysis under vacuum.

| Process Temperature, °C | Rapeseed Oil Processed by Pyrolysis, g | Gases Obtained, g | Pyrolysis Oil Obtained, g | Liquid Product Yield, % |
|---|---|---|---|---|
| 300 | 910.43 | 120.44 | 789.99 | 87% |
| 310 | 872.72 | 337.97 | 534.75 | 61% |
| 320 | 946.23 | 520.43 | 425.80 | 45% |
| 375 | 920.42 | 920.42 | 0 | 0% |

In a vacuum, the pyrolysis oil yield decreases with process temperature increasing, as well as at atmospheric pressure [23] or in the presence of an inert gas [24,25], and in the presence of a catalyst [26,27]. Decreasing of the liquid yield from 87% to 45% over a merely 20 °C range of temperature is influenced by the low pressure strongly favoring the decomposition to gaseous products.

The mineral oils used in quench processes have a low viscosity, this facilitating the thermal transfer and ensuring the temperature decreasing from the core of the piece to the surface in its whole volume. The goal of this study was to obtain bio-oils with lower viscosity in order to improve the heat transfer compared with the raw vegetable oil. The results are shown in Table 3. Also, density, flash points and iodine values of resulting bio-oils are presented in Table 3.

**Table 3.** Influence of pyrolysis temperature on the kinematic viscosity of bio-oils.

| Oil Type | Kinematic Viscosity at 40 °C, mm$^2$/s | Kinematic Viscosity at 100 °C, mm$^2$/s | Density at 20 °C, g/cm$^3$ | Flash Point, °C | Iodine Number, g I$_2$/100 g Sample |
|---|---|---|---|---|---|
| Rapeseed waste oil | 43 | 12.8 | 0.9180 | 202 | 5.8 |
| Bio-oil resulted at 300 °C | 42.9 | 10.9 | 0.9174 | 200 | 5.9 |
| Bio-oil resulted at 310 °C | 42.3 | 9.5 | 0.9162 | 200 | 5.9 |
| Bio-oil resulted at 320 °C | 38.7 | 9.3 | 0.9150 | 198 | 6.0 |
| Mineral oil Iloquench$^{TM}$ 1 | 20 | 5 | 0.8712 | 192 | - |

Even if the viscosity of the bio-oils is higher than the viscosity of the mineral oil at the same temperature, one can see that it decreases with process temperature up to 4.3 units, an important decrease, therefore we expected better performance of this bio-oil in quenching than the performance of raw vegetable oil.

During the thermal decomposition, a dehydration took place as well, thus eliminating the water content which might provoke faults in the structure and even cracks in the material during the quenching process. The iodine number of bio-oils is close to that of the raw (5.8 + max. 0.2 g I$_2$/100 g) sample, in contrast with the bio-oils obtained from the pyrolysis at atmospheric pressure and higher temperature [23–25] in which iodine number increased by up to 2.5 units. This means that, at these relatively low process temperatures, dehydrogenation of the heaviest part in raw oil was minimal, and chemical modifications were minimal in general. The small decrease in iodine number preserves the original stability to oxidation of the bio-oil obtained, this being prone to last as long as the raw, in quenching cycles. The flash points of the bio-oil are also close to the raw, slowly lowering with pyrolysis increasing temperature, so the bio-oil preserved pretty well the quality of the raw. The decrease of the viscosity and density indicates the cracking of longer chains in some molecules in the raw, besides the evaporation of volatiles. The flash point of obtained bio-oils, close to that of raw samples, also indicates the removal of gases and volatile compounds during the process. It should be mentioned that operation at extreme vacuum (in order of $1 \times 10^{-3}$ mbar) is expensive in terms of a large-scale process, but moderate values (5–10 mbar) are feasible in industrial vacuum furnaces. The advantage of such low pressure should be an important decrease in process temperatures, thus leading to lower power consumption in the process versus other methods, such as conventional pyrolysis or the use an inert gas [20,21]. Also, in these conditions, fast pyrolysis takes place, thus shortening the operation time, with consequences for the operating costs.

The quenching capacity of bio-oils is illustrated in Figure 6 showing the IVF Smart Quench apparatus. There are two kinds of curve: cooling curves (temperature vs. time) and cooling rate curves (cooling rate vs. temperature). The different oils are represented in different colors (as seen in Legend).

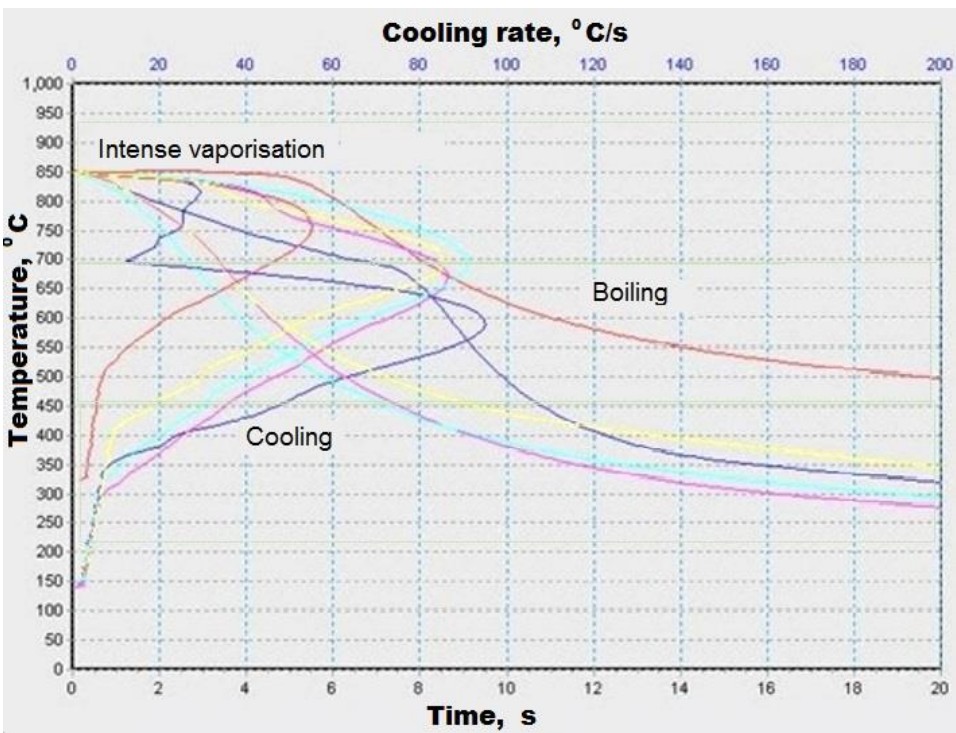

**Figure 6.** Quenching capacity of tested oils; Legend: mineral oil ——— rapeseed oil ——— bio oil at 300 °C ——— bio-oil at 310 °C ——— bio oil at 320 °C ——— .

The cooling curves (temperature vs. time) show a slower cooling in the rapeseed oil and comparable data for the other oils (mineral or bio). The temperature reaches 700 °C in approximately 8 s in the rapeseed oil since in the mineral oil, 700 °C is reached in 6 s and in bio-oils, between 2.5–3.5 s. The cooling rate at 700 °C is superior for bio-oils in (80–90 °C/s) compared with the mineral oil (15 °C/s) and vegetable oil (45 °C/s). A higher cooling rate in the region of 700 °C is better, so the steel perlite transformation is avoided. The cooling rate at 300 °C must be minimized, all the tested oil having close values, between 6 and 8 °C/s, meaning good behavior of cracks and distortions in the material after quenching.
Rockwell harness of probes is shown in Table 4.

**Table 4.** Hardness of stainless steel 25CD4 bars before and after quenching in different oils.

| Oil/Hardness | Before Quenching | After Quenching | | | | |
|---|---|---|---|---|---|---|
| | - | Mineral | Rapeseed Oil | Bio-Oil Obtained at 300 °C | Bio-Oil Obtained at 310 °C | Bio-Oil Obtained at 320 °C |
| HRC units | 29–30 | 45 | 46 | 44–45 | 43 | 43–44 |

The results of quenching are good for every oil/bi-oil used in the process, the Rockwell hardness being improved by 14–16 units. Surprisingly, the rapeseed oil showed the best result even if it was more viscous and had the lowest cooling rate at 700 °C. However, differences in hardness were small, as differences in their characteristics were small (see Table 3), with every oil performing well. Also, there is a great advantage in using the pyrolytic bio-oil since, during the pyrolysis, yields between 13–55% are obtained in products with added value such as gaseous olefins, kerosene-like and diesel-like fractions [23,24] which can be recovered from the vacuum system in the industrial unit.

## 4. Conclusions

This study was designed to find a new means of waste vegetable oil valorisation. The waste frying rapeseed oil was processed by pyrolysis under a vacuum to obtain a bio-oil with reduced viscosity appropriate for use as quenching oil. Such a product was obtained at 1 μbar and 300–320 °C. Its physical–chemical characteristics were close to the raw material, in contrast with the pyrolysis at atmospheric pressure, in the presence or in absence of an inert gas. The kinematic viscosity of this bio-oil was reduced by up to 4.3 mm$^2$/s. The bio-oils performed well in quenching tests on stainless steel probes made from the same material as aircraft pieces, the Rockwell hardness improving by 14–16 units, like the results obtained with the dedicated mineral oil.

An advantage of pyrolysis is that besides the quality bio-oil of 45–87% yield, other valuable products are obtained (gaseous olefins, kerosene-like and diesel-like liquids), resulting in consistent added value to the waste vegetable oil.

**Author Contributions:** Conceptualization, A.M.S., O.V.S. and T.D.; methodology, A.M.S. and O.V.S.; software T.V.C.; validation, D.-R.C.T.; investigation, A.M.S., O.V.S. and A.E.S.; resources A.M.S. and D.-R.C.T.; data curation, O.V.S. and A.E.S.; writing-original draft preparation, O.V.S.; writing-review and editing: T.V.C.; supervision, T.D.; project administration, O.V.S. and T.D.; All authors have read and agreed to the published version of the manuscript.

**Funding:** No funding was received for this study.

**Institutional Review Board Statement:** Not applicable.

**Informed Consent Statement:** Not applicable.

**Data Availability Statement:** All data used to support this study are included within the article.

**Conflicts of Interest:** The authors declare no conflict of interest.

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
