# Peer review of "Thermal Treatment under Vacuum for Obtaining a Quenchant from Rapeseed Oil"

_processes, doi:10.3390/pr9122189_

Round 1

Reviewer 1 Report

This essay studied the fast pyrolysis of rapeseed oil at 300~375 ℃ and 1 μbar absolute pressure. The viscosity, cooling rate curves, cooling curves and 25CD4 hardness before and after quenching of mineral oil, rapeseed oil and bio-oil in the temperature range of 300~320 ℃ are further discussed. It is concluded that the bio-oil prepared by vacuum pyrolysis is a better quenchant. However, there are many problems in this paper, mainly including the following aspects:

  1. In the introduction of the essay, please fully explain the advantages of vacuum pyrolysis bio-oil as a quenching agent compared with mineral oil as a quenching agent, and highlight the significance and novelty of vacuum pyrolysis bio-oil as a quenching agent studied in the article.

  1. In the characteristic column of Table 1, there are two lines of "kinematic viscosity at 40 ℃ mm2/s" in the column of "characteristic". It's puzzling, what does it mean? Is it necessary to modify it to be the same as "kinematic viscosity at 40 ℃ mm2/s" and "kinematic viscosity at 100 ℃ mm2/s" in Table 3 in combination with the introduction? And, in section 2.2 of the essay, the legend in Figure 1 shows a "molecular diffusion pump" at No. 8 and No. 9 respectively. What is the difference and relationship between the two? Please write numbers 10 and 11 separately.

  1. In this paper, only the pyrolysis curve experiment under the conditions of 10-3 mbar pressure, 300 ℃ and 375 ℃ is carried out, and in the abstract which is "preliminary, the raped oil was fast pyrolyzed at temperature in the range of 300~375 ℃ and absolute pressure of 1 μbar". Do you need to supplement the data when the two data cloth match? And the paper also points out that the experimental results at 300 ℃ and 375 ℃ are introduced. How do you choose to do only 310 ℃ and 320 ℃ later?

  1. At the fourth paragraph of section 2.2, “At 330 ℃, the pressure was 1.79·10-3 mbar and at 375 ℃ was 1.97·10-1 mbar and no liquid left in the batch.” The experiment at 330 ℃ and 1.97·10-1 mbar are not introduced in the essay, should they be 300 ℃ and 1.97·10-3 mbar? The format of paragraph 4 of section 2.1 and paragraphs 1-2, 5 and 7-8 of results and discussion are not consistent with the full text. Does "Rockwell Hardness" at paragraph 8 of results and discussion need to be revised to "Rockwell Hardness"?

  1. The last three columns in Table 4, they are "bio-oil obtained at 300 ℃". What is the meaning? Are the last two modified to "bio-oil obtained at 310 °C" and "bio-oil obtained at 320 °C"?

  1. The format of the reference section of the article is not unified, please unify.

Author Response

Dear Reviewer 1,

We thank you again,

The authors

Reviewer 2 Report

The paper reports on the study of the pyrolysis of waste/frying rapeseed oil under vacuum conditions in order to be employed as quenching agent of stainless steel. The manuscript, in the current state has several points that must be improved and, hence, does not have enough quality as there are many questions arising, as it is indicated below:

  • Experimental procedure: it is not specified the amount of raw oil treated in the pyrolysis experiments. In addition, reproducibility of the experiments should be clarified as there is the doubt if oils is lost during vacuum generation. Accordingly, mass balances should be given.
  • In line with the previous comment, the properties of pyrolyzed oils (tabla 3) do not seem to change significantly and are still far from those of the reference mineral oil. A deeper characterization of the pyrolyzed should be done in order to guess the effect of pyrolysis under vacuum. Chemical composition of the oil, such as GC-MS (organic composition), ICP-OES (metals determination)  are examples of analytical techniques that would give useful information. Moreover, the composition of the gases produced could also give information about how the pyrolysis proceeds and help comparing between the different pyrolysis temperatures.
  • Operating under vacuum is quite expensive in terms of a high-scale process, so the benefits in terms of the properties (composition and physico-chemical properties) of the oil obtained should be more clear. If not, this alternative could not be considered relevant enough.
  • As a general comment, the discussion of the results and their relevance should be improved, as well as the quality of the tables and figues (e.g. Fig. 4 is quite difficult to follow).

Author Response

Dear Reviewer 2,

We thank you again,

The authors

Round 2

Reviewer 2 Report

The authors have made an effort to improve the manuscript according to the reviewers suggestions. It could be accepted in the current state, but I encourage the authors to work in the near future to clarify some still unresolved points, such as reproducibility of the experiments or a deep characterization of the oil (by GC-MS) and gases.

Author Response

Dear Reviewer,

Thank you very much for your appreciation.

We will take your advice into account and in the future we will check the reproducibility of the experiment and characterize deeper the products.

In the name of all authors, sincerely yours,

Olga Valerica Săpunaru

Ovidius University of Constanta, Romania
